# Global burden and trends of tracheal, bronchus, and lung cancer attributed to occupational exposure to polycyclic aromatic hydrocarbons in regions with different sociodemographic index, 1990–2021

**Jiansheng Lin, Xiaowei Xie ⑩ \*, Xinyang Zheng, Haizhan Shi**

Department of Thoracic Surgery, The First Hospital of Putian City, Putian, China

\* xxw_biolab@163.com

## Abstract

### Background

Occupational exposure to polycyclic aromatic hydrocarbons (PAHs) is a known risk factor for tracheal, bronchus, and lung (TBL) cancer. However, evidence of its global burden particularly across different socio-demographic index (SDI) regions has been limited.

### Methods

Based on results from the global burden of disease (GBD) study, we conducted a comprehensive analysis of age-standardized death rates (ASDR) and age-standardized disability-adjusted life-years (DALYs) rates due to TBL cancer attributed to occupational exposure to PAHs. This study examined the trends, sex differences, age-specific burden, and regional disparities in TBL cancer burden attributed to occu-pational PAH exposure from 1990 to 2021 globally and across different SDI regions. Age-period-cohort analysis was also performed to evaluate the influence of age, cohort, and period effects.

### Results

Globally, both ASDR and DALYs rates showed slight increases from 1990 to 2021, with estimated annual percentage changes (EAPCs) of 0.76% and 0.54%, respec-tively. Low and middle SDI regions experienced more significant increases in death rates and health burden, while high SDI regions exhibited declining trends. Age-specific analyses revealed higher death rates in older populations, particularly those aged 55–74 years, with rising trends observed in low and middle SDI regions. For high SDI regions, younger age groups (<60 years) showed declining trends, while

**Data availability statement:** The GBD 2021 data used in this study are publicly available online at https://gbd2021.healthdata.org/gbd-results?params=gbd-api-2021-perma-link/8f6fe3b99b879062f3cbfe46014c3935. The minimal anonymized dataset necessary to replicate the study findings is provided in the Supporting information (S1 Data).

**Funding:** The author(s) received no specific funding for this work.

**Competing interests:** The authors have declared that no competing interests exist.

older age groups (>75 years) showed increasing trends. Age-period-cohort analyses indicated that the period effect contributed substantially to rising death rates in low and middle SDI regions, while high SDI regions experienced slower increases in the period effect.

## Conclusions

The study highlights a widening disparity in the burden of TBL cancer due to occupational exposure to PAHs, with lower SDI regions facing greater increases in death rates and DALYs, especially among older populations. Nevertheless, given the inherent limitations of GBD estimation methods and data scarcity in LMICs, the observed disparities should be interpreted with caution and warrant further primary research.

## Introduction

Tracheal, bronchus, and lung (TBL) cancer is one of the leading causes of cancer-related mortality and morbidity worldwide [1,2]. According to the Global Burden of Disease (GBD) study, TBL cancer consistently ranks among the top contributors to cancer-related deaths and burden [3]. TBL cancer poses a significant burden on healthcare systems and economies, particularly in low- and middle-income countries (LMICs) [4]. An increasing number of studies have reported disparities in incidence and survival rates driven by occupational and socio-economic factors [5,6], which underscore the need for a more comprehensive understanding of the occupational factors influencing the TBL cancer burden.

Polycyclic aromatic hydrocarbons (PAHs) are a group of hazardous, lipophilic compounds generated during the incomplete combustion of organic materials [7,8]. PAHs are carcinogens known to increase the risk of TBL cancer [7,9,10]. Occupational exposure to PAHs occurs in industries such as coal mining, petroleum refining, and road paving, where workers are routinely exposed to high concentrations of PAHs [8,11]. Chronic exposure to PAHs has been linked to DNA damage, oxidative stress, and inflammatory responses, all of which contribute to the progression of TBL cancer [7,9,12,13].

Previous studies examining the health burden of TBL cancer have predominantly focused on outdoor air pollution (e.g., emissions from vehicles and industrial activities) and other occupational hazards (e.g., asbestos, silica) [6,14–16]. However, occupational PAH exposure, a significant although overlooked risk factor, remains underexplored at the global scale to date. Related evidence would provide critical insights for occupational health policies aimed at reducing PAH exposure and associated cancer burden.

In addition, socio-economic status significantly influences the health burden of TBL cancer [17,18]. Meanwhile, regions with lower socio-economic development often face higher exposure to environmental and occupational hazards, leading to increased health disparities [18,19]. While existing research highlights the role of socio-economic status in explaining global TBL cancer disparities [17–20], evidence

linking socio-economic status to occupational PAH exposure-related TBL cancer burden remains limited. Addressing this gap is essential for understanding inequities in occupational health and guiding targeted interventions.

This study aimed to provide a detailed evaluation of the global burden of TBL cancer attributable to occupational PAH exposure, particularly across regions with different socio-economic levels, using data from the GBD study. Our study aimed to support the development of evidence-based policies that reduce TBL cancer disparities and improve occupational health globally.

## Methods

### Study design and population

The GBD provides global insights into the prevalence and mortality of diseases, injuries, and risk factors, allowing for comparisons across age groups, sexes, regions, and periods [21]. The national-level study data of GBD can facilitate comparison of health outcomes across countries, enabling researchers and decision-makers to identify leading health challenges and is particularly important for understanding the impact of risk factors [21].

The input data for the GBD study is compiled from a variety of sources, including hospitals, surveys, and government databases. This data is processed and standardized to create consistent, comparable estimates of disease burden across different regions. Regular updates to these estimates ensure that the GBD remains the most accurate and current source of global health data [21]. All data used in this study were publicly available online from the GBD study (https://vizhub. healthdata.org/gbd-compare/).

### Occupational exposure to PAHs and TBL cancer

Data on occupational exposure to PAHs were derived from the GBD study (https://vizhub.healthdata.org/gbd-compare/), with detailed information on data sources and methodology publicly available online (https://www.healthdata.org/gbd/methods-appendices-2021/occupational-risk-factors).

In brief, occupational exposure to PAHs was quantified within the GBD framework using a multi-step process integrating economic activity classifications, occupation distributions, and exposure risk levels [22]. The input data were primarily sourced from the International Labor Organization (ILO). Occupational PAH exposure levels (high/low) were categorized based on the 17 International Standard Industrial Classification (ISIC) economic activities and 9 International Standard Classification of Occupations (ISCO) occupational categories. According to the GBD framework, the Theoretical Minimum Risk Exposure Level (TMREL) of occupational exposure to PAHs was set to zero (no thresholds). High-exposure industries (e.g., coal mining, asphalt production) were assigned elevated exposure rates using the CARcinogen Exposure database and expert-derived thresholds, while low-exposure industries (e.g., education, finance) received lower rates [22,23].

For this study, cancer mortality data were sourced from both cancer registries and the cause of death database. Cancer registry data from 2019 were supplemented with additional data. The inclusion criteria for cancer registries were stringent, focusing on population-based registries that reported data for all cancer types, age groups (excluding pediatric cancers), and both sexes. Priority was given to registries with national coverage, except where the GBD study provides reliable subnational estimates [24,25]. More detailed information is provided online at https://www.healthdata.org/.

### Metrics and variables

In this study, we included death rate and disability-adjusted life years (DALYs) to characterize the health burden of TBL cancer [2,21,23]. Age-standardized death rate (ASDR) represents the death rate attributed to TBL cancer due to occupational exposure to PAHs, adjusted for potential age differences across populations. DALYs is a comprehensive measure of disease burden by combining both years of life lost (YLLs) due to premature death and years lived with disability (YLDs) due to TBL cancer [21]. One DALY is equivalent to one year of healthy life lost, making it a universal metric to compare the

health impact of different diseases across different populations and time frames [21]. Both ASDR and DALYs estimates are reported with 95% uncertainty intervals (UIs), which reflect the certainty of these estimates. The 95% UI is derived by running the estimation model 1,000 times, each time sampling from distributions for data inputs, model choices, and data transformations [21].

The Socio-Demographic Index (SDI) is used to categorize regions with different levels of development in this study [26,27]. SDI combines three factors, income per capita, average years of schooling for individuals aged 15 and older, and the total fertility rate (TFR) for females under age 25 into a single index score ranging from 0 to 100. The data were obtained directly from the GBD study, which aggregates data from standardized international sources, including the World Bank, Demographic Health Surveys, and others. The GBD study provides complete SDI values for all regions and time periods covered in this analysis, with no missing data requiring imputation or adjustment. According to the scores, the SDI regions were categorized into low, low-middle, middle, high-middle, and high SDI regions. Higher SDI values indicate regions with higher levels of socio-economic development [26,27].

## Methodology to calculate attributable burden in the GBD framework

To quantify the disease burden attributable to a specific risk factor, the GBD study used a validated comparative risk assessment (CRA) framework [23]. In brief, the first step is quantifying the relative risks (RRs) of the health outcome (TBL cancer) as a function of exposure to the risk factor (occupational PAHs). This was done using a meta-regression in a "burden of proof" approach, which synthesizes data from systematic reviews. Second, exposure data are projected to the global scale from various sources using Bayesian statistical models, specifically spatiotemporal Gaussian process regression. Third, by integrating projected exposure levels, attributable burden could be calculated to quantify the proportional change in TBL cancer burden that would occur if occupational PAH exposure was reduced to the TMREL. Fourth, the GBD framework accounts for the joint effects of risk factors by assuming that RRs are multiplicative. For risk factors without mediating pathways, such as occupational PAH exposure and confounding factors (e.g., smoking and other occupational carcinogens), their independent contributions were calibrated to avoid overestimation of joint effects [23]. More detailed descriptions of the statistical methods, input data, and exposure-response functions are publicly accessible online (https://ghdx.healthdata.org/record/ihme-data/gbd-2021-burden-by-risk-1990-2021).

## Data collection and processing

Data was gathered from the GBD 2021 result tools. Occupational exposure to PAHs occurs primarily during working-age years (15–75 years). However, due to a prolonged induction-latency period of typically over 10 years from initial exposure to cancer diagnosis [28,29], the attributable mortality burden is observed at older ages. In this study, we collected death and DALYs attributed to TBL cancer due to occupational exposure to PAHs, across specific years (1990–2021), ages (from 25 years to >95 years, at 5-year intervals), sexes, all countries, 21 GBD regions, and different SDI regions. No disease burden was reported for ages <25 years in the GBD 2021 result tools and was not included in the final analyses.

## Statistical analyses

The estimated annual percent change (EAPC) was calculated to quantify the rate of change in ASDR and age-standardized DALYs rate, as well as sex and age-specific death rate and DALYs rate between 1990 and 2021. The EAPC was derived using a log-linear regression model [21,27]. The 95% uncertainty intervals (UIs) for each EAPC were calculated using bootstrapping methods, providing a range of values that reflect the uncertainty in the estimate [21].

The J-point method was used to identify turning points in the temporal trends of TBL cancer burden due to occupational exposure to PAHs, specifically to detect shifts from decreasing to increasing trends (or vice versa) [30,31]. This method involves fitting a series of segmented regression models to identify the point at which the trend of the annual changes most significantly [30,31].

To capture the non-linear association between SDI and ASDR and age-standardized DALYs rate of TBL cancer attributed to occupational exposure to PAHs, we conducted the local polynomial regression (Loess) fit [32,33]. The loess fit does not assume a specific functional form for the relationship between SDI and ASDR/DALYs, allowing for a more flexible representation of the local association between SDI and ASDR and DALYs [32,33].

In addition, to characterize the effects of age, cohort, and period on TBL cancer death rates, an age-period-cohort (APC) model was applied [34,35]. This model helps separate the influence of the age effect (the age at which individuals are with higher risks), the generation or cohort effect (the impact of birth year on risks), and the period effect (the impact of time-specific changes on risks) [34,35]. It is mathematically expressed as $Y_{ijk} = \alpha + f(A_i) + g(P_j) + h(C_k) + \varepsilon_{ijk}$, where $Y_{ijk}$ represents the outcome, α is the intercept, $f(A_i)$ captures the age effect, $g(P_j)$ accounts for period effects, $h(C_k)$ denotes cohort effects, and $\varepsilon_{ijk}$ is the error term. To estimate these effects, the APC model employs the intrinsic estimator method to deal with the collinearity in the cohort constraint equation and allows for robust estimation of independent effects while addressing the identifiability problem [34,35].

All statistical analyses were performed using Stata (version 16) and R (version 4.3.2).

## Results

### Trends of TBL cancer burden attributed to occupational exposure to PAHs in regions with different SDI from 1990 to 2021

As shown in Table 1, globally, the ASDR for TBL cancer due to occupational exposure to PAHs showed a slight increase from 0.05 (95% UI: 0.04, 0.06) per 100,000 in 1990 to 0.07 (95% UI: 0.06, 0.08) per 100,000 in 2021, with an EAPC of 0.76% (95% UI: 0.68, 0.84). The age-standardized DALYs rate increased from 1.65 (95% UI: 1.37, 1.95) per 100,000 in 1990 to 1.98 (95% UI: 1.63, 2.40) per 100,000 in 2021, with an EAPC of 0.54% (95% UI: 0.46, 0.62), indicating an increasing trend in the burden of disease.

The burden of TBL cancer due to occupational exposure to PAHs varied across regions with different SDI levels, with regions having low and middle SDI showing a more significant increase in both ASDR and age-standardized DALYs rate over time. For instance, in low-middle SDI regions, the ASDR increased from 0.07 (95% UI: 0.06, 0.09) per 100,000 in 1990 to 0.11 (95% UI: 0.08, 0.13) per 100,000 in 2021, with an EAPC of 1.28% (95% UI: 1.22, 1.35). In contrast, high SDI regions showed decreasing trends in ASDR and DALYs rate, with EAPCs of −0.72% (95% UI: −0.79, −0.66) and −0.93% (95% UI: −1.00, −0.85), respectively, indicating decreasing trends over the past 30 years.

We also observed sex differences in the burden of TBL cancer attributed to occupational exposure to PAHs. Generally, males exhibited higher ASDR and age-standardized DALYs rates compared to females (S1 Table). However, the EAPCs

**Table 1. Burden and trends of TBL cancer attributed to occupational exposure to PAHs in regions with different SDI, 1990-2021.**

| Region | ASDR (per 100,000) and 95% UI | | | Age-standardized DALYs (per 100,000) and 95% UI | | |
|---|---|---|---|---|---|---|
| | 1990 year | 2021 year | EAPC (%) | 1990 year | 2021 year | EAPC (%) |
| Global | 0.05 (0.04, 0.06) | 0.07 (0.06, 0.08) | 0.76 (0.68, 0.84) | 1.65 (1.37, 1.95) | 1.98 (1.63, 2.40) | 0.54 (0.46, 0.62) |
| Low SDI | 0.04 (0.04, 0.05) | 0.03 (0.03, 0.04) | 0.44 (0.36, 0.52) | 1.23 (1.05, 1.43) | 0.92 (0.79, 1.06) | 0.39 (0.30, 0.47) |
| Low-middle SDI | 0.07 (0.06, 0.09) | 0.11 (0.08, 0.13) | 1.28 (1.22, 1.35) | 2.38 (1.93, 2.85) | 3.07 (2.44, 3.83) | 1.19 (1.13, 1.25) |
| Middle SDI | 0.01 (0.01, 0.02) | 0.01 (0.01, 0.02) | 0.85 (0.78, 0.93) | 0.39 (0.30, 0.54) | 0.46 (0.36, 0.58) | 0.61 (0.53, 0.68) |
| High-middle SDI | 0.02 (0.02, 0.03) | 0.03 (0.03, 0.04) | 1.05 (0.92, 1.18) | 0.69 (0.56, 0.88) | 0.98 (0.81, 1.17) | 0.74 (0.60, 0.88) |
| High SDI | 0.07 (0.06, 0.08) | 0.09 (0.07, 0.12) | −0.72 (−0.79, −0.66) | 2.21 (1.81, 2.65) | 2.71 (2.17, 3.36) | −0.93 (−1.00, −0.85) |

Note: ASDR, age-standardized death rate; DALYs, disability adjusted life-years; EPAC, estimated annual percentage change; SDI, socio-demographic index; UI, uncertainty interval.

for ASDR and age-standardized DALYs rates were higher in females compared to males globally and across regions with different SDI levels, indicating a more apparent increasing trend over time.

Fig 1 further shows the region-specific EAPCs of ASDR and age-standardized DALYs rate of TBL cancer attributed to occupational exposure to PAHs between 1990 and 2021. As for ASDR of both sexes, Solomon Islands, Egypt, and Kenya showed the highest increasing trends, with EAPCs of 4.51% (95% UI: 4.15%, 4.87%), 4.03% (95% UI: 3.52%, 4.54%), 3.58% (95% UI: 3.48%, 3.68%), respectively. Meanwhile, Kazakhstan, Ukraine, and Estonia showed the strongest decreasing trends, with EAPCs of −3.40% (95% UI: −3.55%, −3.25%), −3.34% (95% UI: −3.61%, −3.06%), −3.09% (95% UI: −3.27%, −2.92%), respectively.

Map data were obtained from Resources and Environmental Science Data Platform (public domain, publicly accessible at https://www.resdc.cn/).

Fig 2 shows the annual change in ASDR of TBL cancer attributed to occupational exposure to PAHs in regions with different SDI levels between 1990 and 2021. For both sexes combined, global trends showed a gradual increase in ASDR, although with notable inflection points in 2012 and 2015. Constantly increasing trends were shown in females, while a decreasing trend was shown after 2012 in males. As for disparities across SDI levels, in regions with low SDI, the ASDR increased significantly, especially after 2008. Middle SDI and high-middle SDI regions showed apparent increasing trends after 2015. Conversely, high SDI regions exhibited apparent declines since 1990 and particularly after 2006. For males, middle and high-middle SDI regions showed downward trends after about 2010, and high SDI regions exhibited a decreasing trend since 1990. Nevertheless, for males, only high SDI regions showed a decreasing trend after 2010. The DALYs rate generally showed similar trends, as shown in S1 Fig.

**Age-specific TBL cancer burden attributed to occupational exposure to PAHs in regions with different SDI**

Fig 3 presents age-specific death rates of TBL cancer attributed to occupational exposure to PAHs in regions with different SDI levels for the years 1990 and 2021. For both sexes, the death rates in 2021 showed a clear shift, with higher rates observed in older age groups compared to 1990, particularly in those aged 55–74 years. Fig 4 further shows the age-specific EAPCs of death rates of TBL cancer attributed to occupational exposure to PAHs. For both sexes, regions with low and middle SDI levels showed upward trends in death rates across all age groups, with higher EAPCs observed in older age groups. In contrast, high SDI regions exhibited different trends, with younger age groups (<60 years) showing negative EAPC values, while older age groups (particularly >75 years) showing positive EAPC values (>0). Similar trends were observed for both females and males, with low and middle SDI regions experiencing more significant increases in death rates over time compared to high SDI regions, which showed declining trends in younger populations and increasing trends in older populations.

S2 and S3 Figs also present generally similar trends for age-specific DALYs rate and EAPCs. These results indicated a widening gap across regions with different SDI levels, with low and middle SDI regions experiencing greater increases in death rates and health burden over time, particularly in the older population.

**Associations between SDI and TBL cancer burden attributed to occupational exposure to PAHs in 21 GBD regions**

Figs 5 and S4 show the associations between ASDR and age-standardized DALYs rate of TBL cancer attributed to occupational exposure to PAHs and SDI across different GBD regions from 1990 to 2021. For both sexes combined, we observed an inverted U-shaped relationship between SDI and both ASDR and DALYs rate. Specifically, ASDR and DALYs initially increased with rising SDI (e.g., in East Asia and South Asia), peaking at an SDI value of around 0.6 to 0.7, before declining in high-SDI regions such as North America and Australasia. The strongest increasing trend was observed in East Asia.

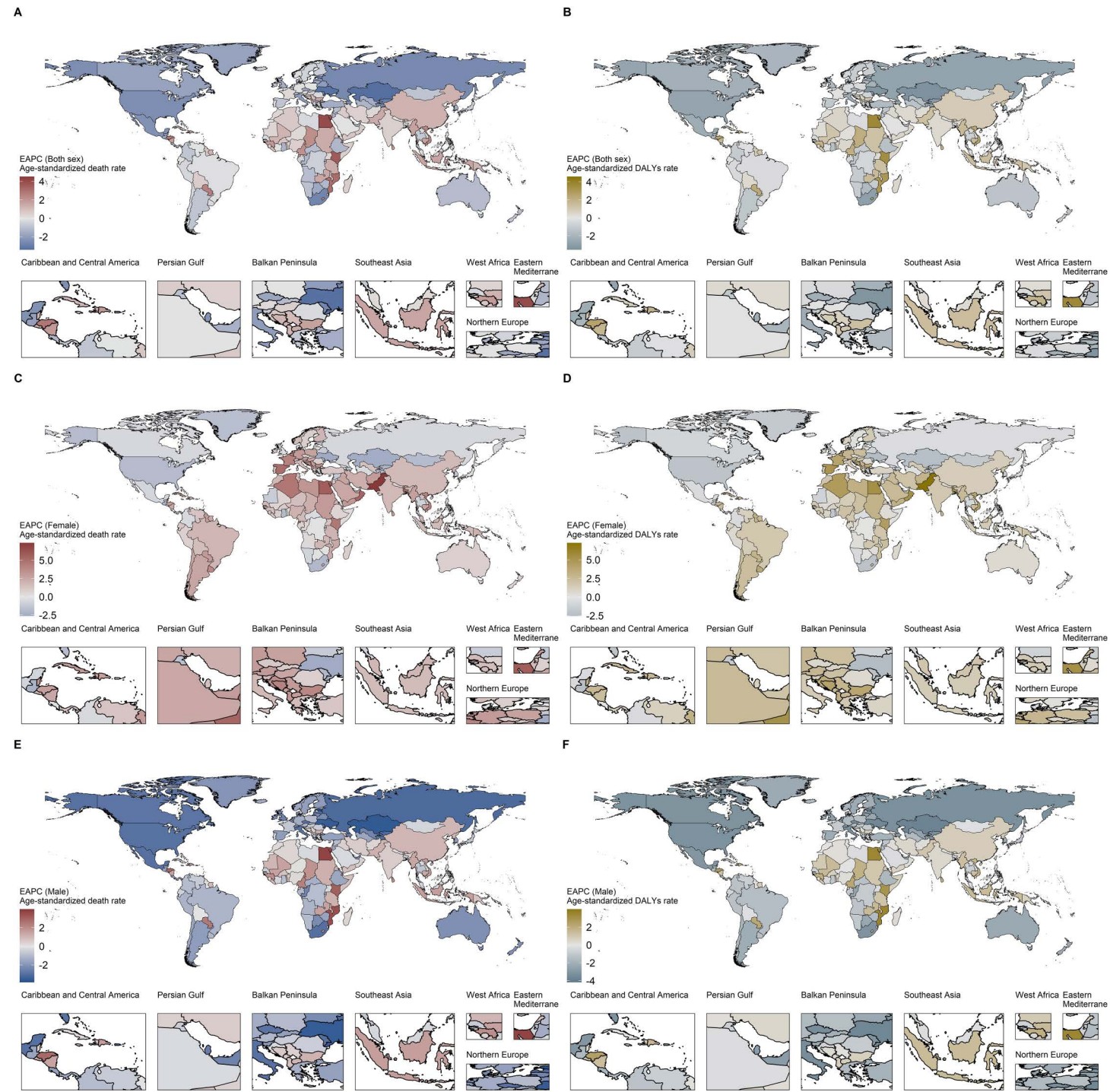

**Fig 1. EAPC of ASDR and age-standardized DALYs rate of TBL cancer attributed to occupational exposure to PAHs from 1990 to 2021. (A)** ASDR for both sex; **(B)** Age-standardized DALYs rate for both sex; **(C)** ASDR for females; **(D)** Age-standardized DALYs rate for females; **(E)** ASDR for males; **(F)** Age-standardized DALYs rate for males. Note: ASDR, age-standardized death rate; DALYs, disability adjusted life-years; EAPC, estimated annual percentage change.

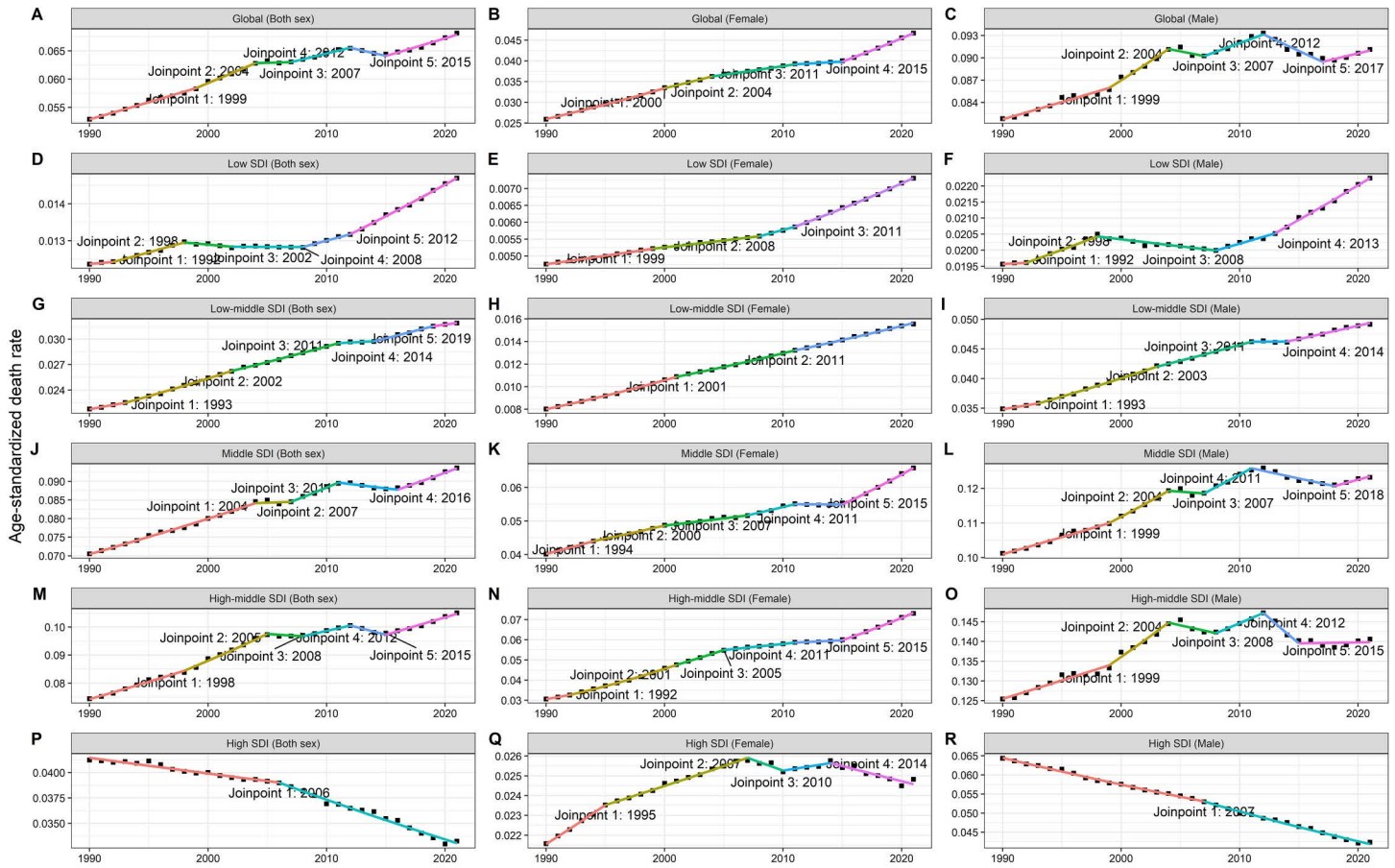

**Fig 2. Annual change in ASDR of TBL cancer attributed to occupational exposure to PAHs in regions with different SDI, 1990-2021.** Note: **(A-C)** Global; **(D-F)** Low SDI; **(G-I)** Low-middle SDI; **(J-L)** Middle SDI; **(M-O)** High-middle SDI; **(P-R)** High SDI. Note: ASDR, age-standardized death rate; SDI, socio-demographic index.

For males, a similar trend was observed, but the peak ASDR occurred at a slightly lower SDI value (approximately 0.6). Notably, declining trends were observed in Southern Latin America and Central Asia, where ASDR decreased. In contrast, for females, the declining trends in high-SDI regions were less pronounced, suggesting a stable burden in these areas.

### APC analysis for TBL cancer burden attributed to occupational exposure to PAHs in regions with different SDI

Fig 6 further shows the effects of age, period, and cohort on the death rates for regions with varying SDI levels from 1990 to 2021. Globally, the ASDR peaked in 55–74 age groups, with an upward trend in the period effect from 1990 to 2021. Notably, the period effect displayed regional variation across different SDI levels. In low and middle SDI regions, the period effect showed a sharp increase, particularly in the most recent years, indicating a rise in death rates linked to occupational exposure to PAHs. In contrast, high SDI regions also exhibited an increasing period effect, but at a slower rate compared to regions with lower SDI. Similar trends were also observed in the APC analyses for DALYs, as shown in S5 Fig.

## Discussion

This study for the first time conducted a comprehensive evaluation of the global burden and trends of TBL cancer attributed to occupational exposure to PAHs from 1990 to 2021, with a focus on disparities across different SDI regions.

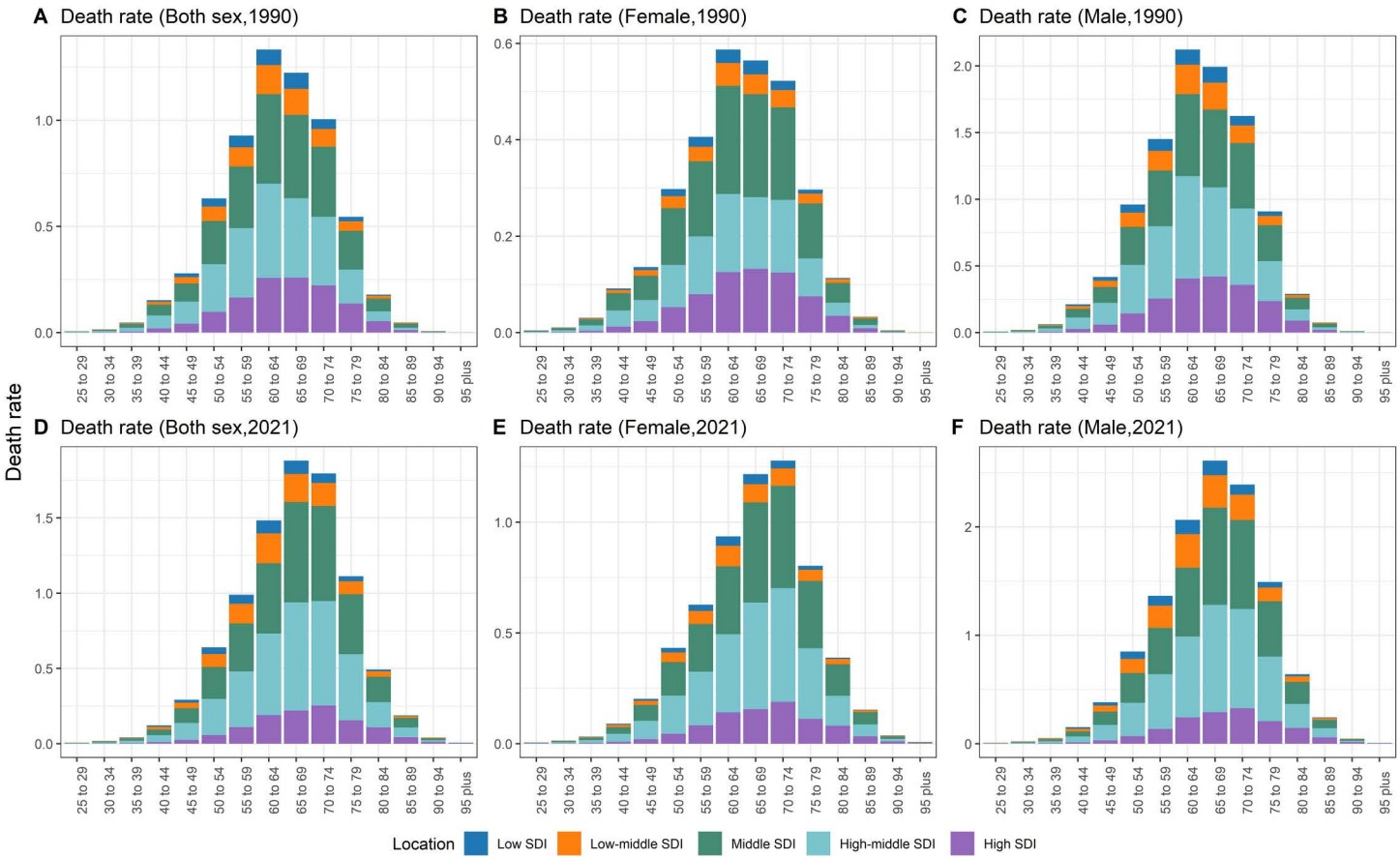

**Fig 3. Age-specific death rate of TBL cancer attributed to occupational exposure to PAHs in regions with different SDI, 1990-2021.** (A-C) 1990; (D-F) 2021. Note: SDI, socio-demographic index.

By examining variations in ASDR, age-standardized DALYs rates, age-specific patterns, and the associations between SDI and TBL cancer burden, our findings provide critical insights into the temporal and regional differences in TBL cancer burden associated with occupational exposure to PAHs. The results highlighted a widening disparity in the burden of TBL cancer due to occupational exposure to PAHs, with lower SDI regions facing greater increases in death rates and DALYs, especially among older populations. By identifying populations at heightened risk, our findings provide key evidence to support targeted occupational health interventions and policymaking efforts aimed at reducing occupational exposure to PAHs and related TBL cancer burden. Addressing these disparities is essential for mitigating inequities in occupational health and advancing global cancer prevention efforts. Nevertheless, it is important to note that the comparisons across SDI regions are based on these modelled estimates, and the observed trends should be interpreted considering potential variability in data quality and completeness underlying the models, particularly in LMICs.

Globally, the ASDR and age-standardized DALYs rate of TBL cancer attributed to occupational PAH exposure have increased slightly over the past three decades. However, the trends vary markedly across SDI levels. Regions with low and middle SDI experienced significant increases in both ASDR and DALYs rates, likely due to higher occupational exposure and limited implementation of preventive measures in these areas [6,36,37]. For instance, the ASDR in low-middle SDI regions increased by 1.28% annually, while high SDI regions showed decreasing trends with an EAPC of −0.72% for ASDR and −0.93% for DALYs rates. Geographic disparities were also observed. Countries such as the Solomon Islands,

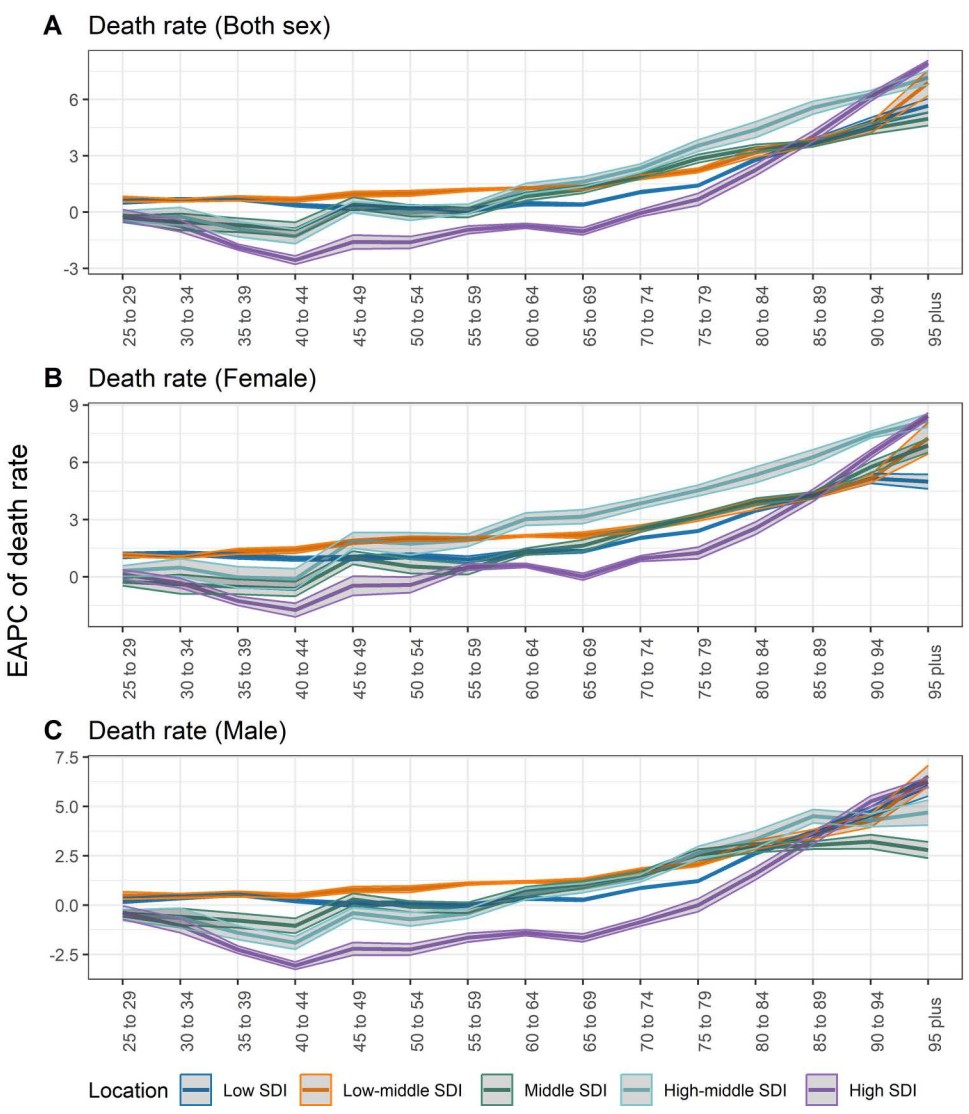

**Fig 4. EAPC of age-specific death rate of TBL cancer attributed to occupational exposure to PAHs in regions with different SDI, 1990-2021. (A)** Both; **(B)** Female; **(C)** Male. Note: EPAC, estimated annual percentage change; SDI, socio-demographic index.

Egypt, and Kenya demonstrated the most pronounced increases in ASDR. These findings emphasize the need for region-specific interventions and policies aimed at mitigating TBL cancer burden associated with PAHs.

Our age-specific analyses revealed a clear trend of increasing death rates of TBL cancer attributed to occupational exposure to PAHs in older age groups, particularly among individuals aged 55–74 years, with notable disparities across regions with different SDI levels. In low and middle SDI regions, this pattern was more pronounced, with rising trends observed across all age groups. This finding could be attributed to several factors. First, older populations in these regions may have experienced prolonged exposure to occupational PAHs due to delayed implementation or absence of strict workplace safety regulations [38–40]. Additionally, limited access to healthcare resources and early detection programs in low and middle SDI regions likely contributes to increasing TBL cancer burden [41,42]. In contrast, in high SDI regions, we observed declining trends among younger age groups (<60 years) and rising trends among older populations (>75 years).

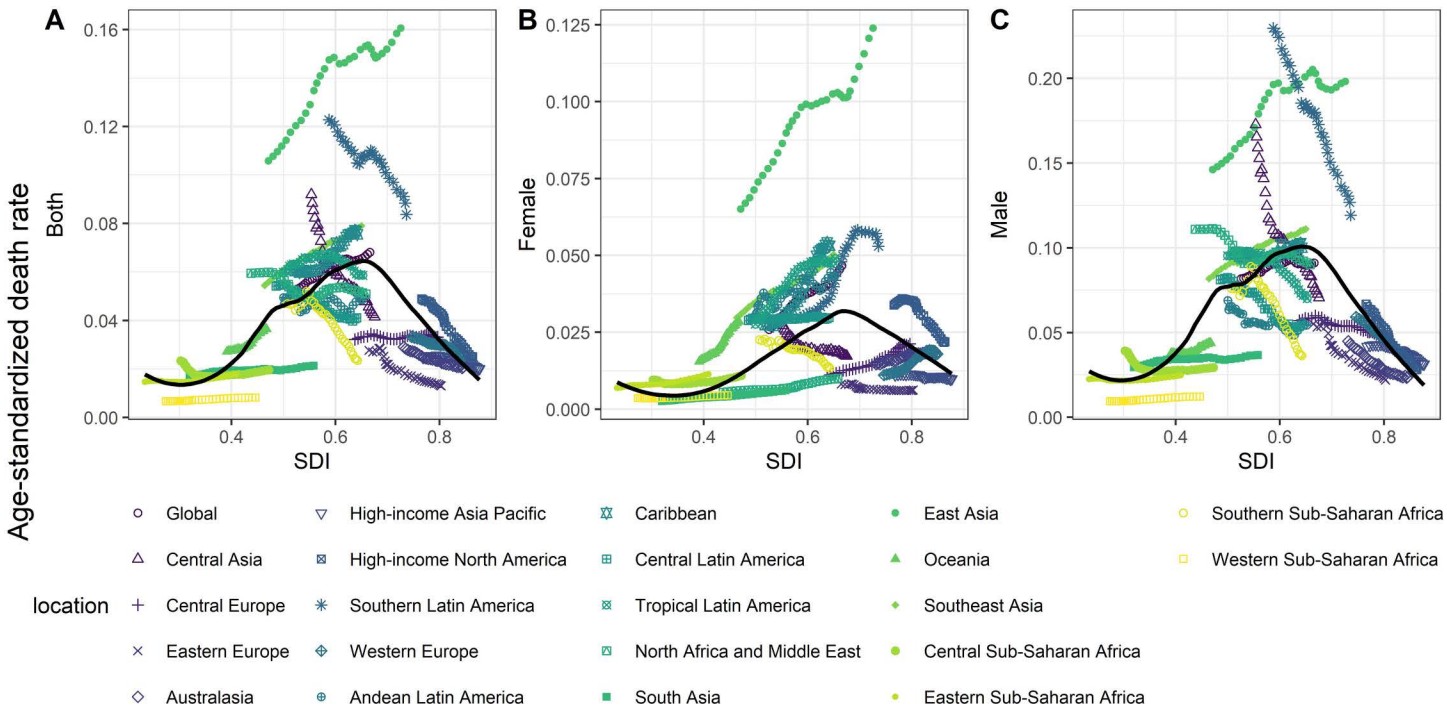

**Fig 5. Associations between ASDR of TBL cancer attributed to occupational exposure to PAHs and SDI in GBD regions, 1990-2021. (A)** Both; **(B)** Female; **(C)** Male. Note: ASDR, age-standardized death rate; SDI, socio-demographic index.

This dichotomy may to some extent reflect historical shifts in occupational exposure, as improved workplace safety regulations and stricter environmental policies in high SDI regions likely reduced PAH exposure for the younger [40,43].

Although males experienced higher ASDRs and DALYs attributable to occupational PAH exposure, females showed larger increasing trends indicated by higher EAPCs. This observed pattern may be driven by a confluence of socioeconomic, occupational, and biological factors. First, shifting occupational demographics play a role. While female workforce participation in formal sectors with potential PAH exposure has increased, gender-based occupational segregation often places women in different roles within these industries (e.g., administrative or support functions versus direct production labor) [22,44]. More critically, in many LMICs, women are disproportionately represented in the informal economy, including waste picking, small-scale food processing using solid fuels, and home-based manufacturing, where high-intensity PAH exposures are common and outside regulatory frameworks [45]. This unregulated exposure likely contributes substantially to the rising trend. Second, the influence of correlated risk factors, particularly active smoking, warrants consideration [10,23,46]. As highlighted by a study among the ten most populous countries, while tobacco-associated lung cancer mortality rates have declined among males in most countries, they have increased among females from 1990 to 2019 [46]. Notably, historical smoking trends have differed by sex, with female smoking prevalence peaking later than male prevalence in many regions [10,47]. Consequently, recent trends in female lung cancer burden likely reflect the combined effects of both risk factors. Although the GBD framework adjusts for smoking independently [23], residual confounding or interaction between smoking and occupational PAHs at the population level may influence the observed sex-specific trends. Third, biological susceptibility may amplify the trends. A previous pooled analysis of 14 case-control studies in Europe and Canada also reported higher risks of lung cancer for ever-exposed women (OR=1.20) versus men (OR=1.08), despite lower median cumulative PAH exposure levels among women [44]. Experimental evidence also showed that women might be more susceptible to PAH-related toxicity due to biological differences. Studies have reported

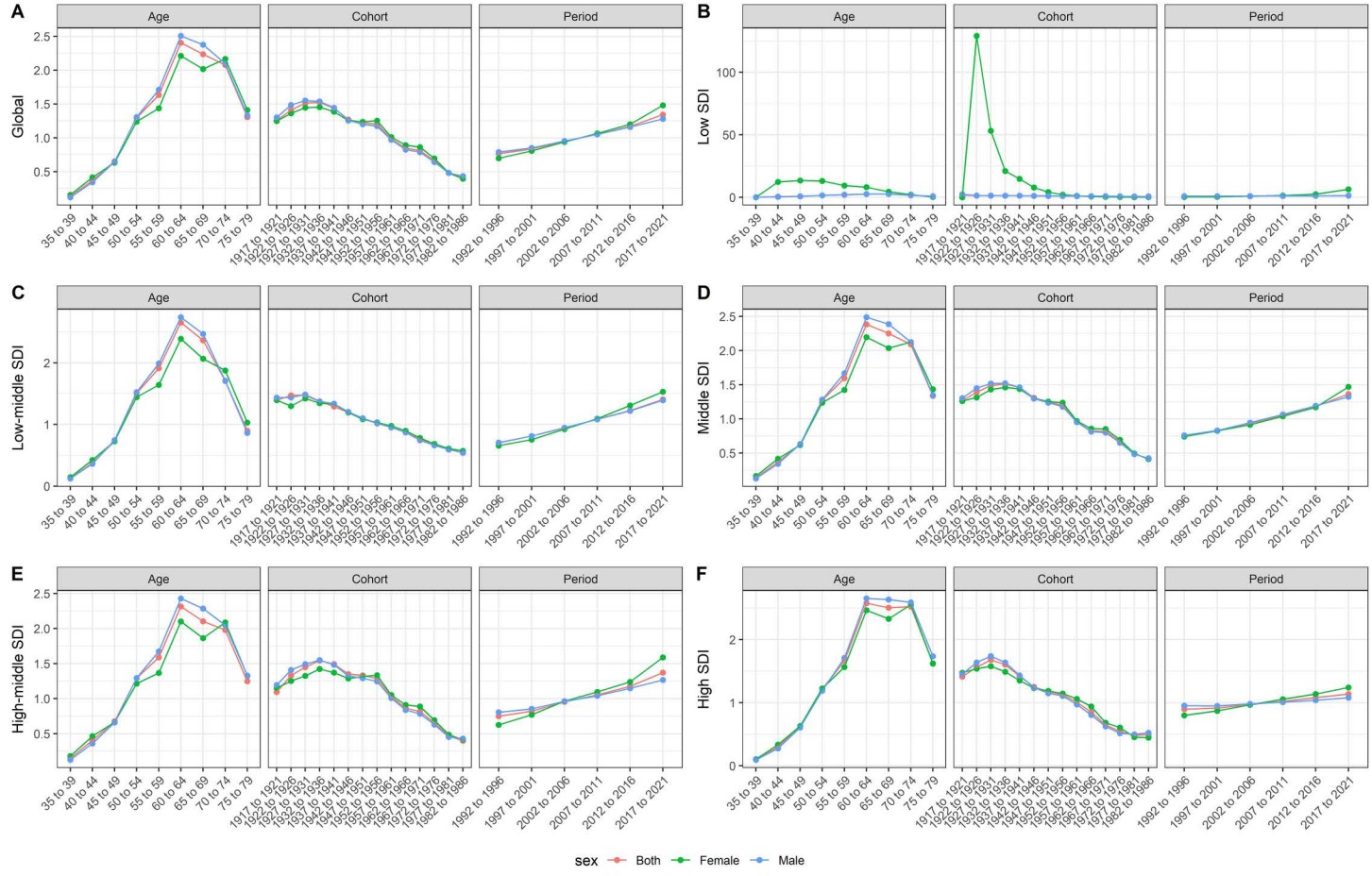

**Fig 6. Age-period-cohort analysis for death rate of TBL cancer attributed to occupational exposure to PAHs. (A)** Global; **(B)** Low SDI; **(C)** Low-middle SDI; **(D)** Middle SDI; **(E)** High-middle SDI; **(F)** High SDI. Note: SDI, socio-demographic index.

higher levels of oxidative stress biomarkers and genotoxic effects in exposed women than in men at similar exposure levels [48]. Furthermore, female lung tissue exhibits higher expression of CYP1A1 and greater accumulation of PAH–DNA adducts, suggesting enhanced metabolic activation of PAHs in women [49]. These biological susceptibilities may amplify the health impact of even modest occupational exposures, contributing to the more significant increasing trends observed in females.

Our analysis identified an inverted U-shaped relationship between SDI and TBL cancer burden, which highlights complex interactions between socioeconomic development and TBL cancer burden associated with occupational exposures to PAHs. The observed inverted U-shaped relationship between SDI and PAH-attributable TBL cancer burden aligns with the framework of the epidemiological transition of occupational risk. In this paradigm, the population-level burden peaks when the pace of industrialization and resultant exposure outstrip the development of protective regulatory frameworks and healthcare capacity. This is evident in low- to middle-SDI regions, where expansion in high-exposure sectors (e.g., manufacturing, construction, informal industry) has driven increased occupational PAH exposure [50,51], while safeguards such as ventilation controls and personal protective equipment remain inadequate. This trend is particularly pronounced in regions such as East Asia, where accelerated industrialization in the late 20th century coincided with the delayed adoption of workplace safety standards [52,53]. The high burden observed today in middle-SDI regions is largely attributable to

ongoing, intensive exposures from current industrial activities. In contrast, the contemporary burden in high-SDI regions primarily stems from exposures accumulated in the past, when these regions were at a similar developmental stage. The declining burden trends now seen in high-SDI settings represent the delayed benefit of occupational health policies, technological improvements, and economic shifts implemented in prior decades [38,40].

The APC analyses further provided additional insights into the temporal dynamics of TBL cancer burden attributed to occupational PAH exposure. It should be noted that the interpretation of cohort patterns is inherently limited by the 30-year study observation window and the multi-decadal latency of TBL cancer, meaning the observed cohort effect integrates exposure experiences over a much longer historical timeframe. Particularly, the strong and increasing period effect in low- and middle-SDI regions quantifies the persistent risk from present-day industrial exposure [39,40,54]. In high-SDI regions, a more modest period effect coupled with elevated risk in older cohorts captures the enduring impact of past exposure regimes [55,56]. Meanwhile, the declining cohort effect observed globally suggests that preventive measures may be reducing risk for more recent generations [56]. Together, these APC findings demonstrate that the inverted U-shaped curve is not merely a cross-sectional disparity but a dynamic signature of industrialization waves, exposure histories, and the delayed translation of TBL cancer burden attributed to occupational PAH exposure.

To reduce the global burden of TBL cancer attributable to occupational PAH exposure, policymakers should prioritize interventions that align with the socio-economic and industrial contexts of regions at different SDI levels. Particularly, in low- and middle-SDI regions, where industrialization often outpaces occupational safety infrastructure, governments should enforce stricter workplace exposure limits for PAHs and the use of personal protective equipment. Efforts should prioritize exposure controls within high-risk industries (e.g., small-scale manufacturing, brick kilns, and informal waste recycling) and practical interventions (e.g., improving natural ventilation, providing access to affordable respiratory protection, and delivering basic occupational safety training) [55]. Globally, harmonizing standards should focus on promoting the ratification and implementation of relevant ILO conventions (https://www.ilo.org/), specifically the Occupational Safety and Health Convention (No. 155) and the Chemicals Convention (No. 170), supported by technical cooperation for monitoring and enforcement. Successful PAH exposure reduction strategies from high-SDI settings, such as the use of enclosed processes and local exhaust ventilation in coke oven operations, offer transferable models that could be adapted to local industrial conditions [55,57]. Furthermore, strengthening local data systems is essential to move beyond reliance on modelled estimates. Establishing linked occupational cancer and exposure registries in sentinel industrial zones of middle-SDI countries would generate locally relevant evidence to directly guide and evaluate policy. Finally, given the rising disease trends among females, integrating gender-specific protections such as targeted safety protocols in female-dominated occupations would also be potentially helpful to mitigate related health burden.

This study is the first comprehensive analysis of the global burden of TBL cancer attributed to occupational PAH exposure, incorporating data from the GBD study across the globe and different SDI regions. Our analyses across SDI regions highlighted critical disparities that can inform targeted interventions. However, several limitations should be noted. First, a significant limitation of this study is its reliance on the modelled estimates of the GBD study. While the GBD employs robust methodologies to synthesize data and address gaps, the accuracy of its estimates is contingent on the quality and coverage of underlying source data. Substantial under-reporting of PAH exposures and TBL cancer diagnoses in many LMICs is a recognized issue, potentially leading to underestimation of the burden [23]. Meanwhile, the ecological nature of the analysis could not fully consider the confounders at the individual level (e.g., tobacco smoking, other occupational carcinogens), which means that the direction and magnitude of net bias are difficult to determine precisely [19]. Therefore, the observed disparities across SDI regions should be interpreted considering potential uncertainties. Second, the lack of more detailed data on exposure duration, intensity, and variability limits our ability to evaluate dose-response relationships comprehensively [22]. Third, differences in healthcare access and diagnostic capabilities across regions and SDI levels may lead to underdiagnosis or misclassification of TBL cancer and may potentially bias the estimates. Fourth, the GBD framework aggregates occupational

exposure across broad industry categories and does not allow further analysis by specific occupations. Consequently, our results could only reflect the average burden among all workers and do not capture the elevated risks faced by the heavily exposed jobs, such as asphalt workers or coke oven workers. Fifth, while the GBD framework used techniques to adjust major risk factors as independent contributors, it could not fully account for potential interactions between occupational PAHs and other exposures, such as outdoor air pollution and co-occurring occupational carcinogens (e.g., asbestos, silica). Sixth, the interpretation of long-term trends in attributable burden can be influenced by changes in other competing risk factors over time. Shifts in the other major risk factors may alter the underlying population at risk for lung cancer, which could affect trends in attributable burden estimates independent of changes in lung cancer burden attributable to occupational PAH exposure.

## Conclusions

The study highlights a widening disparity in the burden of TBL cancer due to occupational exposure to PAHs, with lower SDI regions facing greater increases in death rates and DALYs, especially among older populations. The results underscore the significance of targeted public health interventions in low- and middle-SDI regions to mitigate TBL cancer burden attributed to occupational risks.

## Supporting information

**S1 Table. Burden and trends of TBL cancer attributed to occupational exposure to PAHs in regions with different SDI by sex, 1990–2021.**
(PDF)

**S1 Fig. Annual change in DALYs rate of TBL cancer attributed to occupational exposure to PAHs in regions with different SDI, 1990–2021.** (A-C) Global; (D-F) Low SDI; (G-I) Low-middle SDI; (J-L) Middle SDI; (M-O) High-middle SDI; (P-R) High SDI. Note: DALYs, disability adjusted life-years; SDI, socio-demographic index.
(PDF)

**S2 Fig. Age-specific DALYs rate of TBL cancer attributed to occupational exposure to PAHs in regions with different SDI, 1990–2021.** (A-C) 1990; (D-F) 2021. Note: DALYs, disability adjusted life-years; SDI, socio-demographic index.
(PDF)

**S3 Fig. EAPC of age-specific DALYs rate of TBL cancer attributed to occupational exposure to PAHs in regions with different SDI, 1990–2021.** (A) Both; (B) Female; (C) Male. Note: DALYs, disability adjusted life-years; EPAC, estimated annual percentage change; SDI, socio-demographic index.
(PDF)

**S4 Fig. Associations between age-standardized DALYs rate of TBL cancer attributed to occupational exposure to PAHs and SDI in GBD regions, 1990–2021.** (A) Both; (B) Female; (C) Male. Note: DALYs, disability adjusted life-years; SDI, socio-demographic index.
(PDF)

**S5 Fig. Age-period-cohort analysis for DALYs rate of TBL cancer attributed to occupational exposure to PAHs.** (A) Global; (B) Low SDI; (C) Low-middle SDI; (D) Middle SDI; (E) High-middle SDI; (F) High SDI. Note: DALYs, disability adjusted life-years; SDI, socio-demographic index.
(PDF)

**S1 Data. Minimal anonymized dataset.**
(CSV)

## Acknowledgments

The authors would like to thank the GBD 2021 Collaborators.

## Author contributions

**Conceptualization:** Jiansheng Lin, Xinyang Zheng, Haizhan Shi.

**Formal analysis:** Jiansheng Lin.

**Resources:** Jiansheng Lin, Xinyang Zheng, Haizhan Shi.

**Software:** Jiansheng Lin.

**Supervision:** Xiaowei Xie.

**Writing – original draft:** Jiansheng Lin.

**Writing – review & editing:** Xiaowei Xie.

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
