## [Decision Letter · Decision Letter 0]

2 Aug 2025

Dear Dr. Xie,

We look forward to receiving your revised manuscript.

Kind regards,

Onome Bright Oghenetega, Ph.D.

Academic Editor

PLOS ONE

**Journal Requirements:**

1. When submitting your revision, we need you to address these additional requirements. Please ensure that your manuscript meets PLOS ONE's style requirements, including those for file naming. The PLOS ONE style templates can be found at https://journals.plos.org/plosone/s/file?id=wjVg/PLOSOne_formatting_sample_main_body.pdf and https://journals.plos.org/plosone/s/file?id=ba62/PLOSOne_formatting_sample_title_authors_affiliations.pdf 2. When completing the data availability statement of the submission form, you indicated that you will make your data available on acceptance. We strongly recommend all authors decide on a data sharing plan before acceptance, as the process can be lengthy and hold up publication timelines. Please note that, though access restrictions are acceptable now, your entire data will need to be made freely accessible if your manuscript is accepted for publication. This policy applies to all data except where public deposition would breach compliance with the protocol approved by your research ethics board. If you are unable to adhere to our open data policy, please kindly revise your statement to explain your reasoning and we will seek the editor's input on an exemption. Please be assured that, once you have provided your new statement, the assessment of your exemption will not hold up the peer review process. 3. PLOS requires an ORCID iD for the corresponding author in Editorial Manager on papers submitted after December 6th, 2016. Please ensure that you have an ORCID iD and that it is validated in Editorial Manager. To do this, go to ‘Update my Information’ (in the upper left-hand corner of the main menu), and click on the Fetch/Validate link next to the ORCID field. This will take you to the ORCID site and allow you to create a new iD or authenticate a pre-existing iD in Editorial Manager. 4. We note that Figure 1 in your submission contain map images which may be copyrighted. All PLOS content is published under the Creative Commons Attribution License (CC BY 4.0), which means that the manuscript, images, and Supporting Information files will be freely available online, and any third party is permitted to access, download, copy, distribute, and use these materials in any way, even commercially, with proper attribution. For these reasons, we cannot publish previously copyrighted maps or satellite images created using proprietary data, such as Google software (Google Maps, Street View, and Earth). For more information, see our copyright guidelines: http://journals.plos.org/plosone/s/licenses-and-copyright. We require you to either present written permission from the copyright holder to publish these figures specifically under the CC BY 4.0 license, or remove the figures from your submission: a. You may seek permission from the original copyright holder of Figure 1 to publish the content specifically under the CC BY 4.0 license.   We recommend that you contact the original copyright holder with the Content Permission Form (http://journals.plos.org/plosone/s/file?id=7c09/content-permission-form.pdf) and the following text:“I request permission for the open-access journal PLOS ONE to publish XXX under the Creative Commons Attribution License (CCAL) CC BY 4.0 (http://creativecommons.org/licenses/by/4.0/). Please be aware that this license allows unrestricted use and distribution, even commercially, by third parties. Please reply and provide explicit written permission to publish XXX under a CC BY license and complete the attached form.” Please upload the completed Content Permission Form or other proof of granted permissions as an "Other" file with your submission. In the figure caption of the copyrighted figure, please include the following text: “Reprinted from [ref] under a CC BY license, with permission from [name of publisher], original copyright [original copyright year].” b. If you are unable to obtain permission from the original copyright holder to publish these figures under the CC BY 4.0 license or if the copyright holder’s requirements are incompatible with the CC BY 4.0 license, please either i) remove the figure or ii) supply a replacement figure that complies with the CC BY 4.0 license. Please check copyright information on all replacement figures and update the figure caption with source information. If applicable, please specify in the figure caption text when a figure is similar but not identical to the original image and is therefore for illustrative purposes only.The following resources for replacing copyrighted map figures may be helpful: USGS National Map Viewer (public domain): http://viewer.nationalmap.gov/viewer/The Gateway to Astronaut Photography of Earth (public domain): http://eol.jsc.nasa.gov/sseop/clickmap/Maps at the CIA (public domain): https://www.cia.gov/library/publications/the-world-factbook/index.html and https://www.cia.gov/library/publications/cia-maps-publications/index.htmlNASA Earth Observatory (public domain): http://earthobservatory.nasa.gov/Landsat:
http://landsat.visibleearth.nasa.gov/USGS EROS (Earth Resources Observatory and Science (EROS) Center) (public domain): http://eros.usgs.gov/#Natural Earth (public domain): http://www.naturalearthdata.com/ 5. Please include captions for your Supporting Information files at the end of your manuscript, and update any in-text citations to match accordingly. Please see our Supporting Information guidelines for more information: http://journals.plos.org/plosone/s/supporting-information. 6. If the reviewer comments include a recommendation to cite specific previously published works, please review and evaluate these publications to determine whether they are relevant and should be cited. There is no requirement to cite these works unless the editor has indicated otherwise. 

**Additional Editor Comments:**

Reviewers commend the study but request clearer model assumptions and sensitivity analyses for LMIC data, fuller explanation of GBD exposure attribution, discussion of co-exposure confounding, mechanistic insight into the SDI-disease burden pattern, control of smoking confounding, and rationale for sex differences. They also urge stronger policy and intervention recommendations, especially for low- and middle-SDI regions. These revisions will improve clarity and relevance.

Reviewers' comments:

**Comments to the Author**

1. Is the manuscript technically sound, and do the data support the conclusions?

Reviewer #1: Partly

Reviewer #2: Partly

2. Has the statistical analysis been performed appropriately and rigorously?

Reviewer #1: Yes

Reviewer #2: Yes

3. Have the authors made all data underlying the findings in their manuscript fully available?

Reviewer #1: No

Reviewer #2: Yes

4. Is the manuscript presented in an intelligible fashion and written in standard English?

Reviewer #1: Yes

Reviewer #2: Yes

**Reviewer #1:** 1. The assumptions for the model may be clearly delineated in the methods. Also, the quality of the data from LMICs may be poor, hence sensitivity analyses may be conducted to unravel effect of the assumptions made

2. The manuscript would benefit from more detail on the methodology used by GBD to assign causality between occupational PAH exposure and TBL cancer.

3. Consider briefly discussing confounding by other occupational exposures (e.g., asbestos, silica), which are common co-exposures and may overlap in industrial settings.

4. The finding of an inverted U-shaped relationship between SDI and disease burden is interesting but underexplored. The authors should provide greater mechanistic or contextual explanation—e.g., linking it to patterns in industrialization, occupational regulation, and latency of cancer onset.

5.The paper would benefit from a stronger policy or practical intervention focus, especially in the conclusion. For example, what are the implications for workplace safety standards, surveillance, or global occupational health policy?

**Reviewer #2:**  This study provides a timely and comprehensive analysis of the global burden of tracheal, bronchus, and lung (TBL) cancer attributable to occupational polycyclic aromatic hydrocarbon (PAH) exposure, highlighting disparities across socio-demographic index (SDI) regions from 1990 to 2021. Its strengths include rigorous use of Global Burden of Disease (GBD) data, detailed age-period-cohort analyses, and insightful stratification by SDI, revealing concerning trends in low- and middle-income regions. However, limitations such as reliance on modeled estimates, potential confounding by smoking, and lack of occupational exposure details (e.g., duration, industry-specific risks) warrant clarification. The findings underscore the need for targeted workplace interventions but would benefit from deeper discussion on mechanistic pathways and policy implications to strengthen impact. Overall, the study advances understanding of occupational cancer disparities but requires minor methodological refinements and expanded contextualization for broader relevance.

The study relies on GBD estimates. Could you elaborate on how occupational PAH exposure was specifically measured and attributed to TBL cancer cases in the GBD data? What validation exists for these exposure estimates?

The APC analysis is interesting but complex. Could you provide more details on how the age, period, and cohort effects were separated, and what assumptions were made in this modeling?

You note higher EAPCs in females compared to males globally. What might explain this gender difference, given that occupational PAH exposure has traditionally been higher in male-dominated industries?

The inverted U-shaped relationship between SDI and TBL cancer burden is intriguing. Could you discuss potential mechanisms behind why middle SDI regions show the highest burden?

The manuscript acknowledges uncertainties in GBD estimates. How might underreporting of occupational exposures in low/middle SDI regions affect your findings?

Smoking is a major confounder for lung cancer. How was smoking controlled for in the attribution of TBL cancer to occupational PAH exposure?

Given your findings, what specific interventions would you recommend for low/middle SDI regions to reduce occupational PAH exposure?

How might your results inform international occupational health standards for PAH exposure limits?

Have you considered analyzing specific high-risk occupations (e.g., asphalt workers, coke oven workers) where PAH exposure is particularly high?

Were you able to examine dose-response relationships given the available data?

**Do you want your identity to be public for this peer review?** For information about this choice, including consent withdrawal, please see our Privacy Policy

Reviewer #1: No

Reviewer #2: No

---

## [Author Response · Author response to Decision Letter 1]

5 Aug 2025

Editor

Reviewers commend the study but request clearer model assumptions and sensitivity analyses for LMIC data, fuller explanation of GBD exposure attribution, discussion of co-exposure confounding, mechanistic insight into the SDI-disease burden pattern, control of smoking confounding, and rationale for sex differences. They also urge stronger policy and intervention recommendations, especially for low- and middle-SDI regions. These revisions will improve clarity and relevance.

Response: Thank you for your thoughtful comments and considerations. We have improved our manuscript following your and the reviewers’ comments. The detailed responses to the reviewers are provided in this letter.

Reviewer 1

1. The assumptions for the model may be clearly delineated in the methods. Also, the quality of the data from LMICs may be poor, hence sensitivity analyses may be conducted to unravel effect of the assumptions made

Response: Thank you for this important comment. We agree that a more detailed explanation of the modeling assumptions is crucial. As our study is a secondary analysis of the Global Burden of Disease (GBD) data, the core assumptions and limitations are inherent to the GBD framework itself, and we could not directly calculate the estimates after excluding the original data from LMICs. We have added more content on assumptions for the model and overall quality of the data from LMICs.

In addition, the GBD framework utilizes models like spatiotemporal Gaussian process regression (ST-GPR) to pool this heterogeneous data. These models are crucial for controlling and adjusting for bias and for generating estimates in locations or time periods where direct data may be limited [1]. Nevertheless, it is true that one inherent limitation of the GBD study is the relatively poor data in LMICs.

We have added related content as:

2.4 Methodology to calculate attributable burden in the GBD framework

To quantify the disease burden attributable to a specific risk factor, the GBD study used a validated comparative risk assessment (CRA) framework [1]. In brief, the first step is quantifying the relative risks (RRs) of the health outcome (TBL cancer) as a function of exposure to the risk factor (occupational PAHs). This was done using a meta-regression in a “burden of proof” approach, which synthesizes data from systematic reviews. Second, exposure data are projected to the global scale from various sources using Bayesian statistical models, specifically spatiotemporal Gaussian process regression. Third, by integrating projected exposure levels, attributable burden could be calculated to quantify the proportional change in TBL cancer burden that would occur if occupational PAH exposure was reduced to the TMREL. Fourth, the GBD framework accounts for the joint effects of risk factors by assuming that RRs are multiplicative. For risk factors without mediating pathways, such as occupational PAH exposure and confounding factors (e.g., smoking and other occupational carcinogens), their independent contributions were calibrated to avoid overestimation of joint effects [1]. More detailed descriptions on the statistical methods, input data, and exposure-response functions are publicly accessible online (https://ghdx.healthdata.org/record/ihme-data/gbd-2021-burden-by-risk-1990-2021). Finally, all related estimates were recorded in the GBD 2021 result tools (https://vizhub.healthdata.org/gbd-results/). (Method Section)

First, a key limitation of our study is that the GBD framework is subjected to relatively incomplete and heterogeneous data from LMICs. While the use of spatiotemporal Gaussian process regression helps mitigate data gaps by integrating multiple sources and adjusting for bias, substantial underreporting of occupational exposures and TBL cancer cases in LMICs may lead to underestimations of the true burden [1]. (Discussion Section)

2. The manuscript would benefit from more detail on the methodology used by GBD to assign causality between occupational PAH exposure and TBL cancer.

Response: We agree and appreciate your suggestion, and have added related content as: 2.4 Methodology to calculate attributable burden in the GBD framework

To quantify the disease burden attributable to a specific risk factor, the GBD study used a validated comparative risk assessment (CRA) framework [1]. In brief, the first step is quantifying the relative risks (RRs) of the health outcome (TBL cancer) as a function of exposure to the risk factor (occupational PAHs). This was done using a meta-regression in a “burden of proof” approach, which synthesizes data from systematic reviews. Second, exposure data are projected to the global scale from various sources using Bayesian statistical models, specifically spatiotemporal Gaussian process regression. Third, by integrating projected exposure levels, attributable burden could be calculated to quantify the proportional change in TBL cancer burden that would occur if occupational PAH exposure was reduced to the TMREL. Fourth, the GBD framework accounts for the joint effects of risk factors by assuming that RRs are multiplicative. For risk factors without mediating pathways, such as occupational PAH exposure and confounding factors (e.g., smoking and other occupational carcinogens), their independent contributions were calibrated to avoid overestimation of joint effects [1]. More detailed descriptions on the statistical methods, input data, and exposure-response functions are publicly accessible online (https://ghdx.healthdata.org/record/ihme-data/gbd-2021-burden-by-risk-1990-2021). Finally, all related estimates were recorded in the GBD 2021 result tools (https://vizhub.healthdata.org/gbd-results/). (Method Section)

3. Consider briefly discussing confounding by other occupational exposures (e.g., asbestos, silica), which are common co-exposures and may overlap in industrial settings.

Response: Thank you for this comment. We have added related content: Previous studies examining the health burden of TBL cancer have predominantly focused on outdoor air pollution (e.g., emissions from vehicles and industrial activities) and other occupational hazards (e.g., asbestos, silica) [2-5]. (Introduction Section) & Last, while the GBD framework used techniques to adjust major risk factors as independent contributors, it could not fully account for potential interactions between occupational PAHs and other exposures, such as outdoor air pollution and co-occurring occupational carcinogens (e.g., asbestos, silica). (Discussion Section)

4. The finding of an inverted U-shaped relationship between SDI and disease burden is interesting but underexplored. The authors should provide greater mechanistic or contextual explanation—e.g., linking it to patterns in industrialization, occupational regulation, and latency of cancer onset.

Response: We agree and appreciate your comments. We have added related discussions according to your comments: First, older populations in these regions may have experienced prolonged exposure to occupational PAHs due to delayed implementation or absence of strict workplace safety regulations [6-8]. & In low to middle SDI regions, rapid industrialization particularly in sectors with high PAH exposure risks, such as manufacturing (e.g., aluminum production, coal tar distillation), construction, and informal waste management drives elevated occupational exposure [9, 10]. Meanwhile, inadequate protective measures such as poor ventilation and limited personal protective equipment may also aggravate related disease burden. This trend is particularly pronounced in regions such as East Asia, where accelerated industrialization in the late 20th century coincided with delayed adoption of workplace safety standards [11, 12]. The less apparent decline in females in high SDI regions also suggests potential gender-related disparities in occupational exposure patterns [12]. As regions transition to high SDI, advancements in occupational health policies, automation, and cleaner technologies reduce direct PAH exposure. Stricter regulatory frameworks combined with shifts toward service-oriented economies may also mitigate the TBL cancer burden attributed to occupational exposure to PAHs [6, 8]. (Discussion Section)

5.The paper would benefit from a stronger policy or practical intervention focus, especially in the conclusion. For example, what are the implications for workplace safety standards, surveillance, or global occupational health policy?

Response: Thank you for this important comment. We have related content in Discussion as: To reduce the global burden of TBL cancer associated with occupational PAH exposure, policymakers should prioritize interventions that align with the socio-economic and industrial contexts of regions at different SDI levels. Particularly, in low- and middle-SDI regions, where industrialization often outpaces occupational safety infrastructure, governments should enforce stricter workplace exposure limits for PAHs and the use of personal protective equipment. Globally, harmonizing occupational safety standards through frameworks such as the ILO conventions and promoting technology transfer from high- to low-resource settings could mitigate disparities. Additionally, integrating gender-specific protections such as targeted safety protocols in female-dominated occupations would also be potentially helpful to mitigate related health burden. (Discussion Section)

Reviewer 2

This study provides a timely and comprehensive analysis of the global burden of tracheal, bronchus, and lung (TBL) cancer attributable to occupational polycyclic aromatic hydrocarbon (PAH) exposure, highlighting disparities across socio-demographic index (SDI) regions from 1990 to 2021. Its strengths include rigorous use of Global Burden of Disease (GBD) data, detailed age-period-cohort analyses, and insightful stratification by SDI, revealing concerning trends in low- and middle-income regions. However, limitations such as reliance on modeled estimates, potential confounding by smoking, and lack of occupational exposure details (e.g., duration, industry-specific risks) warrant clarification. The findings underscore the need for targeted workplace interventions but would benefit from deeper discussion on mechanistic pathways and policy implications to strengthen impact. Overall, the study advances understanding of occupational cancer disparities but requires minor methodological refinements and expanded contextualization for broader relevance.

1. The study relies on GBD estimates. Could you elaborate on how occupational PAH exposure was specifically measured and attributed to TBL cancer cases in the GBD data? What validation exists for these exposure estimates?

Response: We appreciate you for this important comment. We have added more information on the details related data we used in the Method Section.

2.2 Occupational exposure to PAHs and TBL cancer

Data on occupational exposure to PAHs were derived from the GBD study (https://vizhub.healthdata.org/gbd-compare/), with detailed information on data sources and methodology publicly available online (https://www.healthdata.org/gbd/methods-appendices-2021/occupational-risk-factors).

In brief, occupational exposure to PAHs was quantified within the GBD framework using a multi-step process integrating economic activity classifications, occupation distributions, and exposure risk levels [13]. The input data were primarily sourced from the International Labor Organization (ILO). Occupational PAH exposure levels (high/low) were categorized based on the 17 International Standard Industrial Classification (ISIC) economic activities and 9 International Standard Classification of Occupations (ISCO) occupational categories. According to the GBD framework, the Theoretical Minimum Risk Exposure Level (TMREL) of occupational exposure to PAHs was set to zero (no thresholds). High-exposure industries (e.g., coal mining, asphalt production) were assigned elevated exposure rates using the CARcinogen Exposure database and expert-derived thresholds, while low-exposure industries (e.g., education, finance) received lower rates [1, 13].

2.4 Methodology to calculate attributable burden in the GBD framework

To quantify the disease burden attributable to a specific risk factor, the GBD study used a validated comparative risk assessment (CRA) framework [1]. In brief, the first step is quantifying the relative risks (RRs) of the health outcome (TBL cancer) as a function of exposure to the risk factor (occupational PAHs). This was done using a meta-regression in a “burden of proof” approach, which synthesizes data from systematic reviews. Second, exposure data are projected to the global scale from various sources using Bayesian statistical models, specifically spatiotemporal Gaussian process regression. Third, by integrating projected exposure levels, attributable burden could be calculated to quantify the proportional change in TBL cancer burden that would occur if occupational PAH exposure was reduced to the TMREL. Fourth, the GBD framework accounts for the joint effects of risk factors by assuming that RRs are multiplicative. For risk factors without mediating pathways, such as occupational PAH exposure and confounding factors (e.g., smoking and other occupational carcinogens), their independent contributions were calibrated to avoid overestimation of joint effects [1]. More detailed descriptions on the statistical methods, input data, and exposure-response functions are publicly accessible online (https://ghdx.healthdata.org/record/ihme-data/gbd-2021-burden-by-risk-1990-2021). Finally, all related estimates were recorded in the GBD 2021 result tools (https://vizhub.healthdata.org/gbd-results/).

2. The APC analysis is interesting but complex. Could you provide more details on how the age, period, and cohort effects were separated, and what assumptions were made in this modeling?

Response: Thank you for this valuable comment. We have added more details on this: In addition, to characterize the effects of age, cohort, and period on TBL cancer death rates, an age-period-cohort (APC) model was applied [14, 15]. This model helps separate the influence of the age effect (the age at which individuals are with higher risks), the generation or cohort effect (the impact of birth year on risks), and the period effect (the impact of time-specific changes on risks) [14, 15]. It is mathematically expressed as Y_ijk=α+f(A_i)+g(P_j)+h(C_k)+ε_ijk, whereY_ijk represents the outcome, α is the intercept, f(A_i) captures the age effect, g(P_j) accounts for period effects, h(C_k) denotes cohort effects, and ε_ijk is the error term. To estimate these effects, the APC model employs the intrinsic estimator method to deal with the collinearity in the cohort constraint equation and allows for robust estimation of independent effects while addressing the identifiability problem [14, 15]. (Method Section)

3. You note higher EAPCs in females compared to males globally. What might explain this gender difference, given that occupational PAH exposure has traditionally been higher in male-dominated industries?

Response: Thank you for this important comment. It is true that although we found males had higher ASDR and DALYs than females, while females presented more significant increasing trends (EAPC). There might be several explanations. In many regions over the past three decades, women’s participation in industries with PAH exposure might have grown [16, 17]. In addition, differences in background lung cancer trends and biological reactions by sex may also amplify the apparent increase in PAH-attributable burden among women [16-18]. We have added potential explanations in the Discussion Section:

Although males experienced higher ASDRs and DALYs attributable to occupational PAH exposure, females showed larger increasing trends indicated by higher EAPCs. This might be driven by several factors. First, over the past three decades, female workforce participation in sectors with incidental PAH exposure has increased in many countries [13, 16]. A previous pooled analysis of 14 case-control studies in Europe and Canada also reported higher risks of lung cancer for ever‑exposed women (OR=1.20) ver

---

## [Decision Letter · Decision Letter 1]

26 Dec 2025

Dear Dr. Xie,

Thank you for submitting your manuscript to PLOS ONE. After careful consideration, we feel that it has merit but does not fully meet PLOS ONE’s publication criteria as it currently stands. Therefore, we invite you to submit a revised version of the manuscript that addresses the points raised during the review process.

We look forward to receiving your revised manuscript.

Kind regards,

Igor Burstyn

Academic Editor

PLOS One

**Journal Requirements:**

Reviewers' comments:

Reviewer's Responses to Questions

**Comments to the Author**

Reviewer #3: All comments have been addressed

Reviewer #4: (No Response)

2. Is the manuscript technically sound, and do the data support the conclusions?

Reviewer #3: Partly

Reviewer #4: Yes

3. Has the statistical analysis been performed appropriately and rigorously?

Reviewer #3: Yes

Reviewer #4: Yes

4. Have the authors made all data underlying the findings in their manuscript fully available?

Reviewer #3: Yes

Reviewer #4: Yes

5. Is the manuscript presented in an intelligible fashion and written in standard English?

Reviewer #3: Yes

Reviewer #4: Yes

**Reviewer #3:**  Major:

1. Over-reliance on GBD Modelling and Data Limitations:

While the authors have expanded the limitations section, the core issue remains that the study's findings are entirely dependent on the GBD's modelled estimates. The acknowledgement of incomplete and heterogeneous data from Low- and Middle-Income Countries (LMICs) is appropriate. However, the statement that findings for lower SDI regions are "conservative estimates" is speculative. The direction of bias (underestimation vs. overestimation) is difficult to ascertain due to competing factors: underreporting of occupational exposures and cancer cases versus potential over-attribution of TBL cancer to PAHs in the absence of robust confounder control at the individual level. This fundamental uncertainty should be more prominently and forcefully stated in the Abstract, Results, and Discussion, framing the entire interpretation of SDI disparities. A sensitivity analysis, though ideal, may not be feasible; therefore, the discussion of this limitation must be exceptionally strong.

2. Superficial Exploration of Gender Differences:

The explanation for higher Estimated Annual Percentage Change (EAPC) in females, while improved, leans heavily on biological susceptibility. This requires more critical balance. The authors should delve deeper into potential socio-occupational factors:

Segregation within industries: Are women increasingly entering specific roles within high-PAH industries that might have different exposure profiles (e.g., administrative roles in manufacturing vs. direct labor)?

Informal sector work: In many LMICs, women are disproportionately represented in informal waste management or small-scale industries where PAH exposure is unmeasured and unregulated. Could this contribute to the trend?

Interaction with other risk factors: The discussion should explicitly consider if trends in female smoking (which vary dramatically by region) could interact with or confound the observed PAH-attributable burden trends. The GBD's adjustment method is described, but its effectiveness in disentangling these correlated risks at the population level warrants a more cautious tone.

3. Mechanistic Explanation of the SDI-Burden Relationship:

The added discussion on the inverted U-shaped curve is helpful but still somewhat descriptive. To strengthen the mechanistic insight:

Link to Economic Transition Theories: Frame the findings within established theories of epidemiological or risk transition. The peak burden in middle-SDI regions mirrors patterns seen with other occupational and environmental hazards, where industrialization outpaces regulatory capacity and health infrastructure.

Latency and Temporal Misalignment: Emphasize that the current burden in high-SDI regions reflects historical exposures from decades past (when their SDI was middle-range). Conversely, the current high burden in middle-SDI regions is a "real-time" consequence of present-day exposures. This temporal disconnect between exposure (past/present) and outcome (present/future) is crucial for interpreting the APC results and forecasting future burdens. The discussion should more explicitly connect the period/cohort effects to these historical industrialization waves.

4. Policy Recommendations Lack Specificity and Feasibility Analysis:

The policy section remains generic. To enhance impact, recommendations should be more targeted and actionable:

Beyond PPE and Limits: While important, recommending "stricter exposure limits and PPE" in low-resource settings is often infeasible without parallel investments in enforcement, monitoring, and worker education. Suggest concrete, incremental steps (e.g., prioritizing exposure control in 2-3 key industries, promoting simple ventilation improvements, developing low-cost exposure biomarkers for surveillance).

Leveraging Existing Frameworks: Mentioning ILO conventions is good. Specify which conventions (e.g., C139, C155, C170) are most relevant and propose a tangible pathway for their adoption and monitoring in target regions.

Case Examples: Briefly reference a successful intervention from a specific country (e.g., reduction in PAH exposures in a particular industry in a high-SDI setting) and discuss its potential for adaptation.

Research-to-Policy Pipeline: Recommend establishing linked occupational cancer registries in sentinel middle-SDI industrial zones to improve data quality and directly inform local policy, moving beyond reliance on global models.

Minor:

1. APC Analysis Interpretation: The interpretation of cohort effects ("recent generations might have lower death rates") is challenging given the 30-year study window and the long latency of TBL cancer. The observed cohort effect likely reflects exposures from the mid-20th century onwards. This complexity should be acknowledged to avoid oversimplification.

2. Figure and Table Presentation:

Table 1: The table is dense. Consider creating a separate, simplified summary table for the main global and SDI-region results in the main text and moving the full sex-stratified table to the supplement.

Figure 1 (Maps): Ensure the color scales are perfectly intuitive and include a clear note in the caption that the maps depict EAPC, not absolute burden.

3. Language and Flow: The manuscript is generally well-written. A final careful edit for concise phrasing and to avoid minor repetitions (e.g., the GBD methodology is described in very similar terms in multiple sections) would enhance readability.

**Reviewer #4:**  Thank you for the opportunity to review your work and I hope you find my comments of assistance. The manuscript addresses an important topic TBL cancer which has an increasing incidence, The attribution to preventable causes such as work is important.

The striking finding for me is the sex differences which show a change in the direction of the usually expected ratios for occupational diseases. These changes in EAPC appear to be marked by country. Noting that the GBD methods for confounders were applied to what extent do the authors feel smoking rates in females may be contributory in comparison to those of males? The paper Evolving trends in lung cancer risk factors in the ten most populous countries: an analysis of data from the 2019 Global Burden of Disease Study Jani, Chinmay T. et al. ClinicalMedicine, Volume 79, 103033 may be helpful although it only examines a limited number of countries. However it is noted, "Among males, tobacco-associated

ASMR fell over time across all countries, excluding China, Indonesia, and Pakistan. Among females, the reverse pattern was observed with overall increasing rates of tobacco-associated ASMR in most countries, with only rates in the USA and Mexico falling between 1990 and 2019".

In addition to the above to what extent is proportional mortality playing a role in survival to die of TBL?

L169 "While occupational exposure primarily occurs during working-age years (typically 15–75 years), the disease burden manifests later in life due to the long latency period of TBL cancer." Figure 3 demonstrates that actually the majority of deaths occur <75 years age. It is perhaps more useful to state the typical latency of TBL cancer.

L49 "Increasing studies have reported disparities in incidence...", suggest "An increasing number of studies have reported disparities in incidence ..."

L55 "PAHs are known carcinogens to increase the risk of TBL cancer" suggest "PAHs are carcinogens known to increase the risk of TBL cancer"

L64 "PAH exposure, a significant while overlooked risk factor" suggest "PAH exposure, a significant although overlooked risk factor"

L446 "subjected to relatively incomplete and heterogeneous data from LMICs." suggest "subject to relatively incomplete and heterogeneous data from LMICs."

Figure 5 is interesting but particularly difficult to distinguish the markers for "Both" and "Female" even when I zoom

**Do you want your identity to be public for this peer review?** For information about this choice, including consent withdrawal, please see our Privacy Policy

Reviewer #3: No

Reviewer #4: No

---

## [Author Response · Author response to Decision Letter 2]

19 Jan 2026

PONE-D-25-19840R1

Global burden and trends of tracheal, bronchus, and lung cancer attributed to occupational exposure to polycyclic aromatic hydrocarbons in regions with different sociodemographic index, 1990-2021

PLOS ONE

Response letter

Dear Dr. Igor Burstyn,

We appreciate your insightful feedback. We have carefully revised the manuscript in accordance with recommendations and believe that these comments have significantly enhanced the manuscript. A point-by-point response to the comments is attached along with the revised manuscript.

We appreciate your guidance and look forward to your response.

Sincerely,

Xiaowei Xie

Department of Thoracic Surgery, The First Hospital of Putian City, Putian, 351100, China

Email: xxw_biolab@163.com

Journal Requirements:

Response: Thank you for your remind. We have carefully reviewed the suggested publication to ensure appropriate citations.

Comments to the Author

1. If the authors have adequately addressed your comments raised in a previous round of review and you feel that this manuscript is now acceptable for publication, you may indicate that here to bypass the “Comments to the Author” section, enter your conflict of interest statement in the “Confidential to Editor” section, and submit your "Accept" recommendation.

Reviewer #3: All comments have been addressed

Reviewer #4: (No Response)

2. Is the manuscript technically sound, and do the data support the conclusions?

Reviewer #3: Partly

Reviewer #4: Yes

3. Has the statistical analysis been performed appropriately and rigorously?

Reviewer #3: Yes

Reviewer #4: Yes

4. Have the authors made all data underlying the findings in their manuscript fully available?

Reviewer #3: Yes

Reviewer #4: Yes

5. Is the manuscript presented in an intelligible fashion and written in standard English?

Reviewer #3: Yes

Reviewer #4: Yes

6. Review Comments to the Author

Reviewer #3:

1. Over-reliance on GBD Modelling and Data Limitations:

While the authors have expanded the limitations section, the core issue remains that the study's findings are entirely dependent on the GBD's modelled estimates. The acknowledgement of incomplete and heterogeneous data from Low- and Middle-Income Countries (LMICs) is appropriate. However, the statement that findings for lower SDI regions are "conservative estimates" is speculative. The direction of bias (underestimation vs. overestimation) is difficult to ascertain due to competing factors: underreporting of occupational exposures and cancer cases versus potential over-attribution of TBL cancer to PAHs in the absence of robust confounder control at the individual level. This fundamental uncertainty should be more prominently and forcefully stated in the Abstract, Results, and Discussion, framing the entire interpretation of SDI disparities. A sensitivity analysis, though ideal, may not be feasible; therefore, the discussion of this limitation must be exceptionally strong.

Response: We would like to thank you for your thoughtful considerations on the limitation of our manuscript. We agree with your comments and have expanded and added related sections.

Revisions:

Abstracts: “Nevertheless, given the inherent limitations of GBD estimation methods and data scarcity in LMICs, the observed disparities should be interpreted with caution and warrant further primary research.”

Results: In the first paragraph of the Discussion Section, after the overview of the results, we have added “Nevertheless, it is important to note that the comparisons across SDI regions are based on these modelled estimates, and the observed trends should be interpreted considering potential variability in data quality and completeness underlying the models, particularly in LMICs.”

Discussion: We have removed the “conservative estimates”, and have added related content: “First, a significant limitation of this study is its reliance on the modelled estimates of the GBD study. While the GBD employs robust methodologies to synthesize data and address gaps, the accuracy of its estimates is contingent on the quality and coverage of underlying source data. Substantial under-reporting of PAH exposures and TBL cancer diagnoses in many LMICs is a recognized issue, potentially leading to underestimation of the burden [1]. Meanwhile, the ecological nature of the analysis could not fully consider the confounders at the individual level (e.g., tobacco smoking, other occupational carcinogens), which means that the direction and magnitude of net bias are difficult to determine precisely [2]. Therefore, the observed disparities across SDI regions should be interpreted considering potential uncertainties.”

2. Superficial Exploration of Gender Differences:

The explanation for higher Estimated Annual Percentage Change (EAPC) in females, while improved, leans heavily on biological susceptibility. This requires more critical balance. The authors should delve deeper into potential socio-occupational factors:

Segregation within industries: Are women increasingly entering specific roles within high-PAH industries that might have different exposure profiles (e.g., administrative roles in manufacturing vs. direct labor)?

Informal sector work: In many LMICs, women are disproportionately represented in informal waste management or small-scale industries where PAH exposure is unmeasured and unregulated. Could this contribute to the trend?

Interaction with other risk factors: The discussion should explicitly consider if trends in female smoking (which vary dramatically by region) could interact with or confound the observed PAH-attributable burden trends. The GBD's adjustment method is described, but its effectiveness in disentangling these correlated risks at the population level warrants a more cautious tone.

Response: We agree with your comments and thank you for this remind. We have improved this section as suggested.

Revision: “Although males experienced higher ASDRs and DALYs attributable to occupational PAH exposure, females showed larger increasing trends indicated by higher EAPCs. This observed pattern may be driven by a confluence of socioeconomic, occupational, and biological factors. First, shifting occupational demographics play a role. While female workforce participation in formal sectors with potential PAH exposure has increased, gender-based occupational segregation often places women in different roles within these industries (e.g., administrative or support functions versus direct production labor) [3, 4]. More critically, in many LMICs, women are disproportionately represented in the informal economy, including waste picking, small-scale food processing using solid fuels, and home-based manufacturing, where high-intensity PAH exposures are common and outside regulatory frameworks [5]. This unregulated exposure likely contributes substantially to the rising trend. Second, the influence of correlated risk factors, particularly active smoking, warrants consideration [1, 6, 7]. As highlighted by a study among the ten most populous countries, while tobacco-associated lung cancer mortality rates have declined among males in most countries, they have increased among females from 1990 to 2019 [7]. Notably, historical smoking trends have differed by sex, with female smoking prevalence peaking later than male prevalence in many regions [6, 8]. Consequently, recent trends in female lung cancer burden likely reflect the combined effects of both risk factors. Although the GBD framework adjusts for smoking independently [1], residual confounding or interaction between smoking and occupational PAHs at the population level may influence the observed sex-specific trends. Third, biological susceptibility may amplify the trends. A previous pooled analysis of 14 case-control studies in Europe and Canada also reported higher risks of lung cancer for ever‑exposed women (OR=1.20) versus men (OR=1.08), despite lower median cumulative PAH exposure levels among women [3]. Experimental evidence also showed that women might be more susceptible to PAH-related toxicity due to biological differences. Studies have reported higher levels of oxidative stress biomarkers and genotoxic effects in exposed women than in men at similar exposure levels [9]. Furthermore, female lung tissue exhibits higher expression of CYP1A1 and greater accumulation of PAH–DNA adducts, suggesting enhanced metabolic activation of PAHs in women [10]. These biological susceptibilities may amplify the health impact of even modest occupational exposures, contributing to the more significant increasing trends observed in females.”

3. Mechanistic Explanation of the SDI-Burden Relationship:

The added discussion on the inverted U-shaped curve is helpful but still somewhat descriptive. To strengthen the mechanistic insight:

Link to Economic Transition Theories: Frame the findings within established theories of epidemiological or risk transition. The peak burden in middle-SDI regions mirrors patterns seen with other occupational and environmental hazards, where industrialization outpaces regulatory capacity and health infrastructure.

Latency and Temporal Misalignment: Emphasize that the current burden in high-SDI regions reflects historical exposures from decades past (when their SDI was middle-range). Conversely, the current high burden in middle-SDI regions is a "real-time" consequence of present-day exposures. This temporal disconnect between exposure (past/present) and outcome (present/future) is crucial for interpreting the APC results and forecasting future burdens. The discussion should more explicitly connect the period/cohort effects to these historical industrialization waves.

Response: Fully agree. We have added this content in revised manuscript.

Revision: “Our analysis identified an inverted U-shaped relationship between SDI and TBL cancer burden, which highlights complex interactions between socioeconomic development and TBL cancer burden associated with occupational exposures to PAHs. The observed inverted U-shaped relationship between SDI and PAH-attributable TBL cancer burden aligns with the framework of the epidemiological transition of occupational risk. In this paradigm, the population-level burden peaks when the pace of industrialization and resultant exposure outstrip the development of protective regulatory frameworks and healthcare capacity. This is evident in low- to middle-SDI regions, where expansion in high-exposure sectors (e.g., manufacturing, construction, informal industry) has driven increased occupational PAH exposure [11, 12], while safeguards such as ventilation controls and personal protective equipment remain inadequate. This trend is particularly pronounced in regions such as East Asia, where accelerated industrialization in the late 20th century coincided with the delayed adoption of workplace safety standards [13, 14]. The high burden observed today in middle-SDI regions is largely attributable to ongoing, intensive exposures from current industrial activities. In contrast, the contemporary burden in high-SDI regions primarily stems from exposures accumulated in the past, when these regions were at a similar developmental stage. The declining burden trends now seen in high-SDI settings represent the delayed benefit of occupational health policies, technological improvements, and economic shifts implemented in prior decades [15, 16].

The APC analyses further provided additional insights into the temporal dynamics of TBL cancer burden attributed to occupational PAH exposure. It should be noted that the interpretation of cohort patterns is inherently limited by the 30-year study observation window and the multi-decadal latency of TBL cancer, meaning the observed cohort effect integrates exposure experiences over a much longer historical timeframe. Particularly, the strong and increasing period effect in low- and middle-SDI regions quantifies the persistent risk from present-day industrial exposure [16-18]. In high-SDI regions, a more modest period effect coupled with elevated risk in older cohorts captures the enduring impact of past exposure regimes [19, 20]. Meanwhile, the declining cohort effect observed globally suggests that preventive measures may be reducing risk for more recent generations [20]. Together, these APC findings demonstrate that the inverted U-shaped curve is not merely a cross-sectional disparity but a dynamic signature of industrialization waves, exposure histories, and the delayed translation of TBL cancer burden attributed to occupational PAH exposure.”

4. Policy Recommendations Lack Specificity and Feasibility Analysis:

The policy section remains generic. To enhance impact, recommendations should be more targeted and actionable:

Beyond PPE and Limits: While important, recommending "stricter exposure limits and PPE" in low-resource settings is often infeasible without parallel investments in enforcement, monitoring, and worker education. Suggest concrete, incremental steps (e.g., prioritizing exposure control in 2-3 key industries, promoting simple ventilation improvements, developing low-cost exposure biomarkers for surveillance).

Leveraging Existing Frameworks: Mentioning ILO conventions is good. Specify which conventions (e.g., C139, C155, C170) are most relevant and propose a tangible pathway for their adoption and monitoring in target regions.

Case Examples: Briefly reference a successful intervention from a specific country (e.g., reduction in PAH exposures in a particular industry in a high-SDI setting) and discuss its potential for adaptation.

Research-to-Policy Pipeline: Recommend establishing linked occupational cancer registries in sentinel middle-SDI industrial zones to improve data quality and directly inform local policy, moving beyond reliance on global models.

Response: We appreciate your comments. According to your comments, we have expanded related content.

Revision: “To reduce the global burden of TBL cancer attributable to occupational PAH exposure, policymakers should prioritize interventions that align with the socio-economic and industrial contexts of regions at different SDI levels. Particularly, in low- and middle-SDI regions, where industrialization often outpaces occupational safety infrastructure, governments should enforce stricter workplace exposure limits for PAHs and the u

---

## [Editor Report · Decision Letter 2]

20 Jan 2026

Global burden and trends of tracheal, bronchus, and lung cancer attributed to occupational exposure to polycyclic aromatic hydrocarbons in regions with different sociodemographic index, 1990-2021

PONE-D-25-19840R2

Dear Dr. Xie,

We’re pleased to inform you that your manuscript has been judged scientifically suitable for publication and will be formally accepted for publication once it meets all outstanding technical requirements.

Kind regards,

Igor Burstyn

Academic Editor

PLOS One

Additional Editor Comments (optional):

Thank you for undertaking revisions. Hopefully you have better appreciation now of the limited utility of GBD data.
---

## [Editor Report · Acceptance letter]

PONE-D-25-19840R2

PLOS One

Dear Dr. Xie,

I'm pleased to inform you that your manuscript has been deemed suitable for publication in PLOS One. Congratulations! Your manuscript is now being handed over to our production team.

Kind regards,

on behalf of

Dr. Igor Burstyn

%CORR_ED_EDITOR_ROLE%

PLOS One